# Demystifying Contrastive Self-Supervised Learning: Invariances, Augmentations and Dataset Biases

**Senthil Purushwalkam**[*]
Carnegie Mellon University
spurushw@cs.cmu.edu

**Abhinav Gupta**
Carnegie Mellon University &
Facebook AI Research
abhinavg@cs.cmu.edu

## Abstract

Self-supervised representation learning approaches have recently surpassed their supervised learning counterparts on downstream tasks like object detection and image classification. Somewhat mysteriously the recent gains in performance come from training instance classification models, treating each image and it's augmented versions as samples of a single class. In this work, we first present quantitative experiments to demystify these gains. We demonstrate that approaches like MOCO[1] and PIRL[2] learn occlusion-invariant representations. However, they fail to capture viewpoint and category instance invariance which are crucial components for object recognition. Second, we demonstrate that these approaches obtain further gains from access to a clean object-centric training dataset like Imagenet. Finally, we propose an approach to leverage unstructured videos to learn representations that possess higher viewpoint invariance. Our results show that the learned representations outperform MOCOv2 trained on the same data in terms of invariances encoded and the performance on downstream image classification and semantic segmentation tasks.

## 1 Introduction

Inspired by biological agents and necessitated by the manual annotation bottleneck, there has been growing interest in self-supervised visual representation learning. Early work in self-supervised learning focused on using "pretext" tasks for which ground-truth is free and can be procured through an automated process [3, 4]. Most pretext tasks include prediction of some hidden portion of input data (e.g., predicting future frames [5] or color of a grayscale image [6]). However, the performance of the learned representations have been far from their supervised counterparts.

The past six months have been revolutionary in the field of self-supervised learning. Several recent works [2, 1, 5, 7, 8] have reported significant improvements in self-supervised learning performance and now surpassing supervised learning seems like a foregone conclusion. So, what has changed dramatically? The common theme across recent works is the focus on the instance discrimination task [9] – treating every instance as a class of its own. The image and its augmentations are positive examples of this class; all other images are treated as negatives. The contrastive loss[5, 7] has proven to be a useful objective function for instance discrimination, but requires gathering pairs of samples belonging to the same class (or instance in this case). To achieve this, all recent works employ an "aggressive" data augmentation strategy where numerous samples can be generated from a single image. Instance discrimination, contrastive loss and aggressive augmentation are the three key ingredients underlying these new gains.

While there have been substantial gains reported on object recognition tasks, the reason behind the gains is still unclear. Our work attempts to demystify these gains and unravel the hidden story behind this success. The utility of a visual representation can be understood by investigating the invariances (see Section 4.1 for definition) it encodes. First, we identify the different invariances that

---

[*]Project webpage: http://www.cs.cmu.edu/~spurushw/publication/demystifyssl/

are crucial for object recognition tasks and then evaluate two state of the art contrastive self-supervised approaches [1, 2] against their supervised counterparts. Our results indicate that a large portion of the recent gains come from occlusion invariances. The occlusion invariance is an obvious byproduct of the aggressive data augmentation which involves cropping and treating small portions of images as belonging to the same class as the full image. When it comes to viewpoint and category instance invariance there is still a gap between the supervised and self-supervised approaches.

Occlusion invariance is a critical attribute for useful representations, but is artificially cropping images the right way to achieve it? The contrastive loss explicitly encourages minimizing the feature distance between positive pairs. In this case, the pair would consist of two possibly non-overlapping cropped regions of an image. For example, in the case of an indoor scene image, one sample could depict a chair and another could depict a table. Here the representation would be forced to be bad at differentiating these chairs and tables - which is intuitively the wrong objective! So why do these approaches work? We hypothesize two possible reasons: (a) The underlying biases of pre-training dataset - Imagenet is an object-centric dataset which ensures that different crops correspond to different parts of same object; (b) the representation function

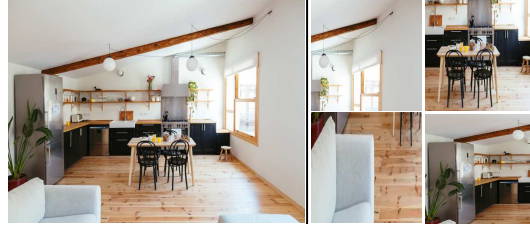

Figure 1: **Aggressive Augmentation** Constrastive self-supervised learning methods employ an aggressive cropping strategy to generate positive pairs. Through this strategy, an image (left) yields many non-overlapping crops (right) as samples. We can observe that the crops do not necessarily depict objects of the same category. Therefore, a representation that matches features of these crops would be detrimental for object recognition tasks.

is not expressive enough to optimize this faulty objective, leading to a sub-optimal representation which works well in practice. We demonstrate through diagnostic experiments that indeed the success of these approaches originates from the object-centric bias of the training dataset. This suggests that the idea of employing aggressive synthetic augmentations must be rethought and improved in future work to ensure scalability.

As a step in this direction, in this paper, we argue for usage of a more natural form of data for the instance discrimination task: videos. We present a simple method for leveraging transformations occurring naturally in videos to learn representations. We demonstrate that leveraging this form of data leads to higher viewpoint invariance when compared to image-based learning. We also show that the learned representation outperforms MoCo-v2 [10] trained on the same data in terms of viewpoint invariance, category instance invariance, occlusion invariance and also demonstrates improved performance on object recognition tasks.

## 2 Contrastive Representation Learning

Contrastive learning [5, 7] is general framework for learning representations that encode similarities according to pre-deteremined criteria. Consider a dataset $\mathcal{D} = \{x_i | x_i \in \mathbb{R}^n, i \in [N]\}$. Let us assume that we have a way to sample positive pairs $(x_i, x_i^+) \in \mathcal{D} \times \mathcal{D}$ for which we desire to have similar representations. We denote the set of all such positive pairs by $\mathcal{D}^+ \subset \mathcal{D} \times \mathcal{D}$. The contrastive learning framework learns a normalized feature embedding $f$ by optimizing the following objective function:

$$\mathcal{L}(D, D^+) = - \sum_{(x,x^+) \in \mathcal{D}^+} \frac{\exp[\, f(x)^\intercal \, f(x^+)/\tau \,]}{\exp[\, f(x)^\intercal \, f(x^+)/\tau \,] + \sum_{\substack{x^- \in \mathcal{D} \\ (x,x^-) \notin D^+}} \exp[\, f(x)^\intercal \, f(x^-)/\tau \,]} \quad (1)$$

Here $\tau$ is a hyperparameter called temperature. The denominator encourages discriminating negative pairs that are not in the positive set $\mathcal{D}^+$. In practice, this summation is expensive to compute for large datasets $\mathcal{D}$ and is performed over $K$ randomly chosen negative pairs for each $x$. Recent works have proposed approaches to scale up the number of negative samples considered while retaining efficiency (see Section 3). In our experiments, we adopt the approach proposed in [10].

The contrastive learning framework relies on the ability to sample positive pairs $(x_i, x_i^+)$. Self-supervised approaches have leveraged a common mechanism: each sample $x$ is transformed using various transformation functions $t \in \mathrm{T}$ to generate new samples. The set of positive pairs is then

considered as $\mathcal{D}^+ = \{(t_i(x), t_j(x)) \mid t_i, t_j \in T, x \in \mathcal{D}\}$ and any pair $(t_i(x), t_k(x'))$ is considered a negative pair if $x \neq x'$.

The choice of transformation functions T controls the properties of the learned representation. Most successful self-supervised approaches [1, 10, 8, 11] have used: 1) cropping sub-regions of images (with areas in the range 20%-100% of the original image), 2) flipping the image horizontally, 3) jittering the color of the image by varying brightness, contrast, saturation and hue, 4) converting to grayscale and 5) applying gaussian blur. By composing these functions and varying their parameters, infinitely many transformations can be constructed.

## 3   Related Work

A large body of research in Computer Vision is dedicated to training feature extraction models, particularly deep neural networks, without the use of human-annotated data. These learned representations are intended to be useful for a wide range of downstream tasks. Research in this domain can be coarsely classified into generative modeling [12, 13, 14, 15, 16, 17] and self-supervised representation learning[3, 4, 18, 19].

**Pretext Tasks** Self-supervised learning involves training deep neural networks by constructing "pretext" tasks for which data can be automatically gathered without human intervention. Numerous such pretext tasks have been proposed in recent literature including predicting relative location of patches in images[3], learning to match tracked patches[4], predicting the angle of rotation in an artificially rotated image[19], predicting the colors in a grayscale image[6] and filling in missing parts of images[20]. These tasks are manually designed by experts to ensure that the learned representations are useful for downstream tasks like object detection, image classification and semantic segmentation. However, the intuitions behind the design are generally not verified experimentally due to the lack of a proper evaluation framework beyond the metrics of the downstream tasks. While we do not study these methods in our work, our proposed framework to understand representations (Section 4) can directly be applied to any representation. In many cases, it can be used to verify the motivations for the pretext tasks.

**Instance Discrimination** Most recent approaches that demonstrate impressive performances on downstream tasks involve training for *Instance Discrimination*. Dating back to [9, 21, 22], the task of instance discrimination involves treating an image and it's transformed versions as one single class. However, the computational costs of performing instance discrimination on large datasets had impeded it's applicability to larger deep neural networks. In NPID[11], the computational expense was avoided using a non-parametric classification method leveraging a memory bank of instance representations. MOCO[1], MOCO-v2[10] adopted the contrastive learning framework(see Section 2) and maintain a queue of negative features which is updated at each iteration. PIRL[2] proposes learning of features which are invariant to the transformations proposed in "pretext" tasks and also uses the memory bank proposed in [11]. At the core, these approaches employ a common mechanism of generating samples for an instance's class - aggressively augmenting the initial image[8, 5, 7, 23].

**SSL from Videos** Self-supervised learning research has also involved leveraging videos for supervision [4, 24, 25, 26]. Specifically, approaches such as [4] and [24] attempt to encode viewpoint and deformation invariances by tracking objects in videos. [26] uses an off-the-shelf motion segmentation as the ground truth for training a segmentation model. Inspired by these works, we propose an approach that tracks regions using weaker self-supervised learning features and uses the tracks to learn better representations within the contrastive learning framework.

**Understanding Self-Supervised Representations** Self-supervised learning methods are evaluated by measuring their performance on numerous downstream tasks[27]. This evaluation framework provides a utilitarian understanding of the representations and fails to provide any insights about why a self-supervised learning approach works for a specific downstream task. There has been some research on developing a more fundamental understanding of the representations learned by deep neural networks in supervised settings [28, 29, 30, 31, 32].

We focus on representations learned by constrastive self-supervised learning methods. In [33], empirical evidence is provided showing that reducing the mutual information between the augmented samples, while keeping task-relevant information intact improves representations. In the context of object recognition, this implies that the category of the augmented sample (task-relevant information) should not change. In our work, we show that the common augmentation methods used in MOCO,

MOCOv2, SimCLR, do not explicitly enforce this and instead rely on a object-centric training dataset bias (see Section 4.2). In [34], the contrastive loss is analyzed to show that it promotes two properties 'alignment' (closeness of features of positive pairs) and 'uniformity' (in the distribution of features on a hypersphere). In our work, we focus on understanding why the learned representations are useful for object recognition tasks. We study two aspects of the representations: 1) invariances encoded in the representations and their relation to the augmentations performed on images and 2) the role of the dataset used for training.

## 4  Demystifying Contrastive SSL

The goal of self-supervised learning in Computer Vision is to learn visual representations. But what is a good visual representation? The current answer [27] seems to be: a representation that is useful for downstream tasks like object detection, image classification, etc. Therefore, self-supervised representations are evaluated by directly measuring the performance on the downstream tasks. However, this only provides a very utilitarian analysis of the the learned representations. It does not provide any feedback as to why an approach works better or insights into the generalization of the representation to other tasks. Most self-supervised learning approaches[4, 3, 1, 10] provide intuitions and conjectures for the efficacy of the learned representations. However, in order to systematically understand and improve self-supervised learning methods, a more fundamental analysis of these representations is essential.

### 4.1  Measuring Invariances

Invariance to transformations is a crucial component of representations in order to be deployable in downstream tasks. A representation function $h(x)$ defined on domain $\mathcal{X}$ is said to be invariant to a transformation $t : \mathcal{X} \to \mathcal{X}$ if $h(t(x)) = h(x)$. An important question to ask is what invariances do we need?

An ideal representation would be invariant to all the transformations that do not change the target/ground-truth label for a task. Consider a ground-truth labeling mechanism $y = \mathsf{Y}(x)$ (where $x \in \mathcal{X}, y \in \mathcal{Y}$ such that $\mathcal{Y}$ is the set of all labels). An ideal representation $h^*(x)$ would be invariant to all the transformations $t : \mathcal{X} \to \mathcal{X}$ that do not change the target i.e. if $\mathsf{Y}(t(x)) = \mathsf{Y}(x)$, then $h^*(t(x)) = h^*(x)$. In object recognition tasks, a few important transformations that do not change the target are viewpoint change, deformations, illumination change, occlusion and category instance invariance. We seek representations that do not change too much when these factors are varied for the same object.

We formulate an approach to measure task-relevant invariances in representations. We adopt the approach proposed in [28] with some modifications to incorporate dependence on the task labels. Consider a representation $h(x) \in R^n$ where each dimension is the output of a hidden unit. According to [28], the $i$-th hidden unit is said to fire when $s_i h_i(x) > t_i$ where the threshold $t_i$ is chosen according to a heuristic described next and $s_i \in \{-1, 1\}$ allows a hidden unit to use either low or high activation values to fire. For each hidden unit, $s_i$ is selected to maximize the considered invariance. Using this definition, a *firing representation* $f(x) \in R^n$ can be constructed where each dimension is the indicator of the corresponding hidden unit firing *i.e.* $f_i(x) = \mathbb{1}(s_i h_i(x) > t_i)$.

The *global firing rate* of each hidden unit is defined as $G(i) = \mathbb{E}(f_i(x))$. This is controlled by the chosen threshold $t_i$. In this work, we choose the thresholds such that $G(i) = 1/|\mathcal{Y}|$. Intuitively, we choose a threshold such that the number of samples the hidden unit fires on is equal to (or close to) the number of samples in each class[2].

A *local trajectory* $T(x) = \{t(x, \gamma) \mid \forall \gamma\}$ is a set a transformed versions of a reference input $x \in \mathcal{X}$ under the parametric transformation $t$. For example, for measuring viewpoint invariance, $T(x)$ would contain different viewpoints of $x$. The *local firing rate* for target $y$, is defined as:

$$L_y(i) = \frac{1}{|\mathcal{X}_y|} \sum_{z \in \mathcal{X}_y} \frac{1}{|T(z)|} \sum_{x \in T(z)} f_i(x) \quad \text{where} \quad \mathcal{X}_y = \{x | x \in \mathcal{X}, \mathsf{Y}(x) = y\} \qquad (2)$$

**Table 1: Invariances learned from Imagenet:** We compare invariances encoded in supervised and self-supervised representations learned on the Imagenet dataset. We consider invariances that are useful for object recognition tasks. See text for details about the datasets used. We observe that compared to the supervised model, the contrastive self-supervised approaches are better only at occlusion invariance.

| Dataset | Method | Occlusion | | Viewpoint | | Illumination Dir. | | Illumination Color | | Instance | | Instance+Viewpoint | |
|---|---|---|---|---|---|---|---|---|---|---|---|---|---|
| | | Top-10 | Top-25 | Top-10 | Top-25 | Top-10 | Top-25 | Top-10 | Top-25 | Top-10 | Top-25 | Top-10 | Top-25 |
| Imagenet | Sup. R50 | 80.89 | 74.21 | 89.54 | 82.62 | 94.63 | 89.08 | 99.88 | 99.38 | 66.11 | 59.44 | 70.17 | 63.47 |
| Imagenet | MOCOv2 | 84.19 | 77.88 | 85.15 | 75.08 | 90.28 | 80.76 | 99.66 | 97.11 | 62.49 | 55.01 | 67.4 | 60.52 |
| Imagenet | PIRL | 84.46 | 78.38 | 85.8 | 76.08 | 87.7 | 78.45 | 99.68 | 97.19 | 52.97 | 46.79 | 57.01 | 51.03 |

Intuitively, $L_y(i)$ measures the fraction of transformed inputs (of target $y$) on which the $i$-th neuron fires. Normalizing the local firing rate by the global firing rate gives us the *target conditioned invariance* for the $i$-th hidden unit as $I_y(i) = \frac{L_y(i)}{G(i)}$.

The final *Top-K Representation Invariance Score (RIS)* can be computed by averaging target conditioned invariance for top-K neurons (selected to maximize RIS) and computing the mean over all targets. We convert the Top-K RIS to a percentage of the maximum possible value (i.e. for all neurons $L_y(i) = 1 \ \forall y \in \mathcal{Y}$). For discussion on differences from [28], please see supplementary material Appendix A.

We can now investigate the invariances encoded in the constrastive self-supervised representations and their dependence on the training data. Since we wish to study the properties relevant for object recognition tasks, we focus on invariances to viewpoint, occlusion, illumination direction, illumination color, instance and a combination of instance and viewpoint changes. We now describe the datasets used to evaluate these invariances and will publicly release the code to reproduce the invariance evaluation metrics on these datasets.

**Occlusion**: We use the training set of the GOT-10K tracking dataset[35] which consists of videos, every frame annotated with object bounding boxes and the amount of occlusion (0-100% occlusion discretized into 8 bins). We crop each bounding box to create a separate image. *Local trajectories* consisting of varying occlusions are constructed for each video by using one sample for each unique level of occlusion.

**Viewpoint+Instance and Instance** We use the PASCAL3D+ dataset[36] which consists of images depicting objects from 12 categories, annotated with bounding boxes and the viewpoint angle with respect to reference CAD models. We again crop each bounding box to create a separate image. *Local trajectories* consisting of objects from the same category, but different viewpoints are collected by ensuring that each trajectory only contains one image for each unique viewpoint. Additionally, we can construct local trajectories containing objects belonging to the same category and depicted in the same viewpoint, restricting the transformation to instance changes only.

**Viewpoint, Illumination Direction and Illumination Color** The ALOI dataset[37] contains images of 1000 objects taken on a turntable by varying viewpoint, illumination direction and illumination color separately. Therefore, the dataset directly provides 1000 local trajectories for each of the annotated properties.

**Discussion** The aggressive cropping in MOCO and PIRL creates pairs of images that depict parts of objects, thereby simulating occluded objects. Therefore, learning to match features of these pairs should induce occlusion invariance. From our results, we do observe that the self-supervised approaches MOCO and PIRL have significantly higher occlusion invariance compared to an Imagenet supervised model. PIRL has slightly better occlusion invariance compared to MOCO which be attributed to the more aggressive cropping transformation used by PIRL. However, the self-supervised approaches are inferior at capturing viewpoint invariance, and significantly inferior at instance and instance+viewpoint invariance. This can be attributed to the fact that instance discrimination explicitly forces the self-supervised models to minimize instance invariance. We also observe that the ImageNet supervised model possesses higher Illumination Color Invariance compared to the ImageNet-MOCO model. At first glance, this might seem counter-intuitive since MOCO employs an aggressive color augmentation strategy unlike the ImageNet model. However, we believe that this can be attributed to the difference in the color changes depicted in the augmentation and in the local trajectories considered for measuring this invariance. The natural illumination color changes for measuring invariances are restricted to illumination ***temperature*** changes from 2175K to 3075K[37] - this is also naturally captured in the ImageNet dataset. The synthetically color-jitterred samples are significantly different from these images depicting arbitrary color changes

## 4.2 Augmentation and Dataset Biases

The results above raise an interesting question: how do self-supervised approaches outperform even supervised approaches on occlusion invariances. As discussed above, the answer lies in how contrastive self-supervised learning construct positive examples. Most approaches treat random crops (from 20% to 100% of original image) of images as the positive pairs which essentially is matching features of partially visible (or occluded) images. Note that PIRL[2] follows an even more agressive strategy: dividing a random crop further into a 3x3 grid.

But this aggressive augmentation comes at a cost. Consider the example of an indoor scene shown in the Figure 1(left) and the random crops shown in Figure 1(right). Contrastive learning on such positive pairs effectively forces the couch, dining table, refridgerator and the window to have similar representations. Such a representation is clearly not beneficial for object discriminating tasks. However, the learned approaches still demonstrate strong results for image classification. We hypothesize that this could be due to *dataset biases*: the pre-training and downstream datasets are biased in an advantageous manner.

**Biases:** Contrastive self-supervised approaches are most commonly trained on the ImageNet dataset. Images in this dataset have an object-centric bias: single object is depicted, generally in the center of the image. This dataset bias is highly advantageous for constrastive self-supervised learning approaches since the random crops always include a portion of an object and not include objects from other categories. While PIRL [2] has also used YFCC[38] which are less biased, the evaluation framework does not effectively evaluate the discriminative power. For example, in image classification, if test images very frequently contain both couches and television, representations that do not differentiate them can still achieve seemingly impressive performances. Furthermore, background features are generally strongly tied with the objects depicted. We believe that these biases exist in the standard classification benchmark - Pascal VOC[39].

In order to verify the hypothesis of pre-training dataset bias, we first construct a new pre-training and downstream image classification task. We pretrain self-supervised models on the MSCOCO dataset[40] which is more scene-centric and does not suffer the object-centric bias like Imagenet. Instead of using the standard VOC classification benchmark for evaluation, we crop the annotated bounding boxes in this dataset to include only one object per image (referred to as **Pascal Cropped Boxes**). This allows us to focus on the model's discriminative power.

**Table 2: Discriminative power of representations:** We compare representations trained on different datasets, in supervised and self-supervised settings, on the task of image classification. We observe that representations trained on object-centric datasets, like Imagenet and cropped boxes from MSCOCO, are better at discriminating objects. We also demonstrate that the standard classification setting of Pascal VOC is not an ideal testbed for self-supervised representations since it does not test the ability to discriminate frequently co-occurring objects.

| Dataset | Method | Pascal<br>Mean AP | Pascal Cropped Boxes<br>Mean AP | ImageNet<br>Top-1 Acc |
|---|---|---|---|---|
| ImageNet | Supervised | 87.5 | 90.13 | 76.5 |
| ImageNet | MOCOv2 | 83.3 | 90.03 | 67.5 |
| ImageNet | PIRL | 81.1 | 84.82 | 63.6 |
| ImageNet 10% | MOCOv2 | 62.32 | 73.85 | 38.53 |
| MSCOCO | MOCOv2 | 64.39 | 71.94 | 33.64 |
| MSCOCO Boxes | MOCOv2 | 59.6 | 75.29 | 34.24 |

In this experiment, we train three MOCOv2 models: trained on 118K MSCOCO images, trained on a randomly sampled 10% subset of ImageNet (similar number of images as MSCOCO) and trained on a dataset of 118K cropped bounding boxes from the MSCOCO dataset. The results are shown in Table 2. We observe that MOCOv2 trained on MSCOCO outperforms the model trained on MSCOCO Boxes on the standard Pascal dataset (Column 1). This could be due to two reasons: 1) due to the co-occurrence and background biases of Pascal (discussed above) which is favorable for models trained on full MSCOCO images or 2) MSCOCO Cropped boxes represent a significantly smaller number of pixels and diversity of samples compared to the full MSCOCO. On the other hand, the trend is reversed when tested on Pascal cropped boxes (Column 2). In this setting, the MOCOv2 model trained on full COCO images cannot rely on co-occurrence statistics and background. However, The object-centric bias of the MSCOCO cropped boxes leads to higher discrimination power. A similar trend is observed in comparison to the MOCOv2 model trained on the Imagenet 10% (which

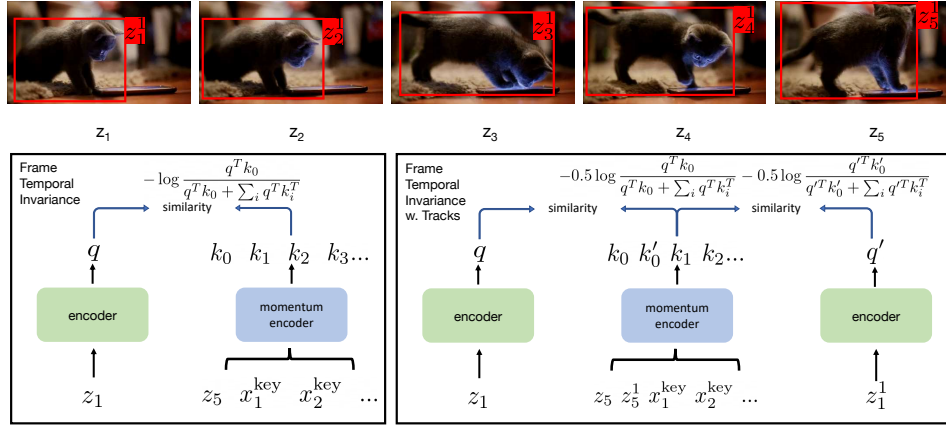

**Figure 2: Leveraging Temporal Transformations:** We propose an approach to leverage the naturally occurring transformations in videos and learn representations in the MOCOv2 framework. The Frame Temporal Invariance model uses full frames and tracked region proposals separated in time as the query and key. See supplementary material Appendix B for additional implementation details.

also possesses a strong object-centric bias) [3]. This indicates that the aggressive cropping is harmful in object discrimination and does not lead to right representation learning objective unless trained on an object-centric dataset.

# 5 Learning from Videos

Since our analysis suggests that aggressive cropping is detrimental, we aim to explore an alternative in order to improve the visual representation learned by MOCOv2. Specifically, we would like to focus on improving invariance to viewpoint and deformation since they are not captured by the MOCOv2 augmentation strategy. One obvious source of data is videos since objects naturally undergo deformations, viewpoint changes, illumination changes and are frequently occluded. We refer to these transformations collectively as *Temporal Transformations*. Since we seek representations that are invariant to these transformations, such videos provide the ideal training data. Consider the dataset of videos $v \in \mathcal{V}$ where each video $v = (v_i)_{i=1}^{\mathsf{N}(v)}$ is a sequence of $\mathsf{N}(v)$ frames.

**Baseline** The naive approach for learning representations from this dataset would be to consider the set of all frames $\{z_i | z \in \mathcal{V}, i \in \mathsf{N}(z)\}$ and apply a self-supervised contrastive learning method. We evaluate this baseline by training MOCOv2 on frames extracted from TrackingNet[41] videos. Note that in practice we extract 3 frames per video uniformly spaced apart in time.

**Frame Temporal Invariance** The baseline approach ignores the natural transformations occurring in videos. We therefore propose an alternative approach to leverage these temporal transformations to learn viewpoint invariant representations. We first construct a dataset of pairs of frames: $\mathcal{V}_{\text{pairs}} = \{(z_i, z_{i+k}) \mid z \in \mathcal{V}, i \in \mathsf{N}(z), i \bmod k = 0\}$. In each of these pairs, $z_{i+k}$ captures a naturally transformed version of $z_i$ and vice versa. For training under contrastive learning, we can create a set of 118K positive pairs by applying the standard transformations on these frames separately. Following the notation in Section 2, $\mathcal{D}^+ = \{(t_i(z_i), t_j(z_{i+k})) \mid t_i, t_j \in \mathrm{T}, (z_i, z_{i+\Delta}) \in \mathcal{V}_{\text{pairs}}\}$ where $\mathrm{T}$ is the set of transformations used in MOCO-v2.

While this captures the temporal transformations occurring in the frames, the learned features focus on frame-level scene representations. In order to be effective for object recognition tasks, we desire representations that encode *objects* robustly. As also demonstrated in Section 4.2, training on images that are not object-centric decreases the robustness of the representations. Therefore, we propose an extension to the Frame Temporal Invariance model.

**Region Tracker** Each frame $z_i$ is further divided into $R$ regions $\{z_i^r\}_{r=1}^R$ using an off-the-shelf unsupervised region proposal method[42]. In order to find temporally transformed versions of each region, we track the region in time through the video. This is done by matching each region $z_i^r$ to a region in a subsequent frame $z_{i+\Delta}^s$ by choosing the minimum distance between the region features *i.e.* $s = \arg\min_{r'} d(z_i^r, z_{i+\Delta}^{r'})$. While any unsupervised feature representation can be used for this, we use the baseline model described above and pool features at `layer3` of the ResNet using ROI-Pooling[43]. By recursively matching regions between $\{z_i^r\}_{r=1}^R, \{z_{i+\Delta}^r\}_{r=1}^R, \{z_{i+2\Delta}^r\}_{r=1}^R, ...$ and thresholding, we can generate tracks of the form $(z_i^r, z_{i+k}^s)$. These tracks can be used as positive pairs for contrastive learning. We employ a similar training approach as the Frame Temporal Invariance model, but with an additional contrastive loss to match positive region pairs and discriminate negative region pairs. We provide more concrete implementation details in the supplementary material.

**Table 3: Evaluating Video representations:** We evaluate our proposed approach to learn representations by leveraging *temporal transformations* in the contrastive learning framework. We observe that leveraging frame-level and region-level temporal transformations improves the discriminative power of the representations. We present results on four datasets - Pascal, Pascal Cropped Boxes, Imagenet (image classification) and ADE20K (semantic segmentation).

| Dataset | Pascal<br>Mean AP | Pascal Cropped Boxes<br>Mean AP | ImageNet<br>Top-1 | ADE20K<br>Mean IOU | ADE20K<br>Pixel Acc. |
|---|---|---|---|---|---|
| Baseline MOCOv2 | 61.8 | 70.91 | 30.33 | 14.69 | 61.78 |
| Frame Temp. Invariance | 63.89 | 72.17 | 29.34 | 14.41 | 61.85 |
| Ground Truth Tracks | 66.21 | 76.16 | 37.45 | 14.69 | 61.78 |
| Region Tracker | 66.47 | 75.86 | 36.51 | 15.28 | 63.29 |

**Table 4: Invariances of Video representations:** We evaluate the invariances in the representations learned by our proposed approach that leverages frame-level (row 2) and region-level (row 3, 4) temporal transformations. We observe compared to the Baseline MOCOv2 model, the models that leverage temporal transformations demonstrate higher viewpoint invariance, illumination invariance, category instance invariance and instance+viewpoint invariance.

| Method | Occlusion | | Viewpoint | | Illumination Dir. | | Illumination Col. | | Instance | | Instance+Viewpoint | |
|---|---|---|---|---|---|---|---|---|---|---|---|---|
| | Top-10 | Top-25 | Top-10 | Top-25 | Top-10 | Top-25 | Top-10 | Top-25 | Top-10 | Top-25 | Top-10 | Top-25 |
| Baseline MOCOv2 | 81.73 | 75.35 | 81.55 | 71.71 | 82.19 | 72.45 | 98.78 | 93.58 | 43.76 | 40.43 | 48.85 | 45.76 |
| Frame Temp. Invariance | 79.92 | 73.33 | 83.87 | 74.86 | 84.47 | 75.57 | 99.18 | 96.03 | 42.98 | 39.42 | 47.81 | 44.26 |
| Ground Truth Tracks | 81.52 | 74.6 | 84.82 | 75.3 | 88.28 | 78.51 | 99.92 | 98.31 | 47.51 | 42.93 | 53.47 | 48.63 |
| Region Tracker | 83.26 | 76.52 | 84.97 | 76.18 | 88.3 | 79.34 | 99.77 | 97.7 | 48.81 | 44.38 | 53.31 | 49.04 |
| Imagenet 10% MOCOv2 | 84 | 78.26 | 80.42 | 70.42 | 81.9 | 72.27 | 98.29 | 92.71 | 46.23 | 42.65 | 48.54 | 45.46 |
| Imagenet MOCOv2 | 84.19 | 77.88 | 85.15 | 75.08 | 90.28 | 80.76 | 99.66 | 97.11 | 62.49 | 55.01 | 67.4 | 60.52 |

We now evaluate the representations learned from videos using the proposed approaches. First, we perform a quantitative evaluation of the approaches on downstream tasks. We then analyze the invariances learned in this representation by following the framework established in Section 4.1.

## 5.1 Evaluating Temporal Invariance Models

We evaluate the learned representations for the task of image classification by training a Linear SVMs (for Pascal, Pascal Cropped boxes) and a linear softmax classifier (for Imagenet). We also evaluate on the task of semantic segmentation on ADE20K[44] by training a two-layered upsampling neural network[45]. In Table 3, we report the evaluation metrics to compare the three models presented in Section 5. The Ground Truth tracks model uses annotated tracks rather than unsupervised tracks. We observe that the Frame Temporal Invariance representation outperforms the Baseline MOCO model on the Pascal classification tasks. We additionally observe that the Region-Tracker achieves the best performance on these all tasks demonstrating stronger discriminative power.

## 5.2 Analyzing Temporal Invariance Models

The Frame Temporal Invariance and Region-Tracker representations were explicitly trained to be robust to the naturally occurring transformations in videos. Intuitively, we expect these representations to have higher viewpoint invariance compared to the Baseline MOCO. In Table 4, we report the Top-K RIS percentages for the three representations. Our analysis confirms that the two proposed representations indeed have significantly higher viewpoint invariance. Most importantly, we observe that the Region-Tracker model has significantly higher viewpoint and illumination dir. invarance compared to MOCOv2 trained on a 10% subset of Imagenet (same number of samples) and is comparable to the MOCOv2 model trained on full Imagenet (10x the number of samples).

# 6 Conclusion

The goal of this work is to demystify the efficacy of constrastive self-supervised representations on object recognition tasks. We present a framework to evaluate invariances in representations. Using this framework, we demonstrate that these self-supervised representations learn occlusion invariance by employing an aggressive cropping strategy which heavily relies on an object-centric dataset bias. We also demonstrate that compared to supervised models, these representations possess inferior viewpoint, illumination direction and category instance invariances. Finally, we propose an alternative strategy to improve invariances in these representations by leveraging naturally occurring temporal transformations in videos.

## Broader Impact

The goal of this work is to analyze existing self-supervised learning methods through diagnostic experiments. Analysis and understanding of existing approaches help develop better interpretation of ML algorithms and can be crucial in removing biases. Upon identifying the shortcomings of existing approaches, we propose a modification to improve the representations learned by these approaches. Self-supervised learning involves learning representations from a large collection of unlabeled data. Since there is no human involvement in the data collection pipeline, we anticipate reduction in biases that can come via human labeling. Furthermore, self-supervised learning is a relatively nascent research topic with minimal deployability in the real-world. Therefore, while in the long run visual self-supervised learning would be impactful, at this moment there is no immediate impact.

## Acknowledgments and Disclosure of Funding

CMU effort was supported by ONR MURI, DARPA MCS, SAIL-ON and ONR Young Investigator Award. We would like to thank Ishan Misra & Shubham Tulsiani for the useful discussion, and feedback on the paper.

## Footnotes

[2]Note that this heuristic is only applicable for datasets with uniformly distributed targets and has been presented to simplify notation. See supplementary material Appendix A for a more general formulation of this heuristic.

[3]This could also be explained by the train-test domain gap *i.e.* full scene images vs cropped boxes. In order to discredit this explanation, we create a separate test-dataset consisting of the subset of Pascal VOC07 test images which depict either table or chair in the image, but not both *i.e.* full images containing only one of a frequently co-occurring pair of objects. We observe that on the table vs chair full image classification task, the representation trained on COCO-Boxes (74.92mAP) outperforms full COCO-image (73.64mAP) pre-training.

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
