[Supplementary Material]

# (Supplementary Material) Demystifying Contrastive Self-Supervised Learning: Invariances, Augmentations and Dataset Biases

## A    Comparison of Invariance Measure to Goodfellow et. al[28]

In Section 4.1 of the main text, we presented an approach to measure invariances in representations. This approach was directly adopted from [28] with some minor modifications. In this section, we describe these differences and the motivation for these modifications.

In our work, we wish to measure invariances encoded in representations while accounting for the discriminative power of the representations. However, in [28], the focus is purely on measuring invariances which in many cases could assign higher scores to representations that are not discriminative. This is manifested in the following changes:

- **Chosen Thresholds** In [28], the threshold for each hidden unit is chosen to be a constant such that the global firing rate is 0.01 (i.e. the hidden unit fires on 1% of all samples). In contrast, in our work, we choose an adaptive threshold for each class in the dataset. For a specific class $y$, we choose the threshold such that the global firing rate is $G_y(i) = P(y)$ (i.e. the fraction of samples having label $y$). This allows each hidden unit the ability to fire on all samples having class $y$. In contrast, the threshold chosen in [28] could lead to a hidden unit firing on only a fraction of the samples of class $y$ (if $p(y) > 0.01$). Consider a hidden unit that consistently has higher activations for samples of class $y$. Such a hidden unit is optimally invariant and discriminative, by could have lower invariance scores under the heuristic of [28] when a local trajectory contains a higher-scoring and a lower-scoring sample of $y$. Note that the heuristic presented in the main paper for simplicity of notation is only applicable for datasets with uniform distribution of labels where $G_y(i) = P(y) = 1/|\mathcal{Y}|$.

- **Local Firing Rate** Since in our work we choose thresholds that are class-dependent, we need to compute separate local firing rates considering the local trajectories for each class $L_y(i)$. This has the added benefit of assigning equal importance to samples of each class, especially in class-imbalanced datasets. This is in contrast to [28], where a single local firing rate is computed across all local trajectories of all classes (denoted by $L(i)$ in [28]). This assigns higher weights to classes with larger number of samples, hence disregarding the discriminative power of representations.

- **Invariance Scores** Since in our work we compute class-dependent local firing rates, we first compute task-dependent invariance scores $I_y(i) = L_y(i)/G_y(i)$. The Top-K hidden units are chosen for each class separately and the mean task-dependent invariance score is computed.

$$I(f) = \frac{1}{|\mathcal{Y}|} \sum_{y \in \mathcal{Y}} \frac{1}{K} \Big[ \max_{\substack{|C|=K \\ C \subseteq [n]}} \sum_{i \in C} I_y(i) \Big] \tag{3}$$

In [28], the Top-K hidden units are chosen across all classes, again penalizing hidden units that are optimally discriminative and invariant for specific classes.

We believe that these modifications are essential to measure invariances in representations that are intended to be used in tasks that require discrimination of classes.

## B    Implementation Details: Learning from Videos

In Section 5 of the main text, we present an approach to leverage naturally occurring *temporal transformations* to train models in the MOCOv2 framework[10]. In Algorithm 1, we provide pseudo-code to allow reproducibility of this method. In this section, we also describe the dataset creation, unsupervised tracking method and other implementation details.

**Dataset Creation**    For experiments in Section 5, we use the TrackingNet dataset[41] that consists of 30K video sequences. In order to increase the size of the dataset, from each video we extract 4 temporal chunks of 60 consecutive frames such that the chunks are maximally spaced apart in time. Each chunk is considered a separate video for all training purposes.

**Algorithm 1** MoCo-style Pseudo-code for Frame Temporal Invariance.

```
# f_q, f_k: encoder networks for query and key                                            1
# queue: dictionary as a queue of K keys (CxK)                                            2
# m: momentum                                                                             3
# t: temperature                                                                          4
# use_tracks: True for Frame Temporal Invariance with tracks                              5
                                                                                          6
def get_loss_and_keys(x1, x2):                                                            7
    x_q = aug(x1) # a randomly augmented version                                          8
    x_k = aug(x2) # another randomly augmented version                                    9
    q = f_q.forward(x_q) # queries: NxC                                                  10
    k = f_k.forward(x_k) # keys: NxC                                                     11
    k = k.detach() # no gradient to keys                                                 12
    # positive logits: Nx1                                                               13
    l_pos = bmm(q.view(N,1,C), k.view(N,C,1))                                            14
    # negative logits: NxK                                                               15
    l_neg = mm(q.view(N,C), queue.view(C,K))                                             16
    # logits: Nx(1+K)                                                                    17
    logits = cat([l_pos, l_neg], dim=1)                                                  18
    # contrastive loss, Eqn.(1)                                                          19
    labels = zeros(N) # positives are the 0-th                                           20
    loss = CrossEntropyLoss(logits/t, labels)                                            21
    return loss, k                                                                       22
                                                                                         23
f_k.params = f_q.params # initialize                                                     24
for x1, x2 in loader: # load a minibatch of frame pairs x1, x2 with N samples            25
    loss, k = get_loss_and_keys(x1, x2)                                                  26
                                                                                         27
    if use_tracks:                                                                       28
        x1_patch, x2_patch = sample_track(x1, x2) # Sample a patch pair tracked from frame x1 to frame x2   29
        loss_patch, k_patch = get_loss_and_keys(x1_patch, x2_patch)                      30
        loss = 0.5*loss + 0.5*loss_patch                                                 31
                                                                                         32
    # SGD update: query network                                                          33
    loss.backward()                                                                      34
    update(f_q.params)                                                                   35
                                                                                         36
    # momentum update: key network                                                       37
    f_k.params = m*f_k.params+(1-m)*f_q.params                                           38
                                                                                         39
    # update dictionary                                                                  40
    enqueue(queue, k) # enqueue the current minibatch                                    41
    dequeue(queue) # dequeue the earliest minibatch                                      42
                                                                                         43
    if use_tracks:                                                                       44
        enqueue(queue, k_patch)                                                          45
        dequeue(queue)                                                                   46
```

bmm: batch matrix multiplication; mm: matrix multiplication; cat: concatenation.

**Generating Tracks**    For each frame, we extract region proposals using the unsupervised method - selective search[42]. We choose the top 300 region proposals for frames which produce more than 300 regions. Following the notation from the main text, each video $v = (v_i)_{i=1}^{N(v)}$ is a sequence of $N(v)$ frames. Each frame consists of $R$ regions $\{z_i^r\}_{r=1}^R$. The matching score between region $z_i^r$ and a region $z_j^{r'}$ is defined as the cosine similarity between their features $f$ i.e. $\max(0, d_{\cos}(f(z_i^r), f(z_j^{r'})))$. Here the features $f(x)$ are extracted by ROI-pooling the layer 3 of the ResNet model $f$. The score of a track from region $z_i^r$ to region $z_j^{r'}$ is defined using the following recursive expression:

$$S(z_i^r, z_j^{r'}) = \sum_k \frac{j-i-1}{j-i} S(z_i^r, z_{j-1}^k) * \max(0, d_{\cos}(f(z_{j-1}^k), f(z_j^{r'}))) \tag{4}$$

$$S(z_t^r, z_{t+1}^k) = \max(0, d_{\cos}(f(z_t^r), f(z_{t+1}^k))) \ \forall r, t, k \tag{5}$$

For any pair of frames, we only consider tracks that have a score above a chosen threshold.

**Sampling Frames**    Training the Frame Temporal Invariance model requires sampling pairs of frames that are temporally separated $\mathcal{V}_{\text{pairs}} = \{(z_i, z_{i+k}) \mid z \in \mathcal{V}, i \in \mathsf{N}(z), i \bmod k = 0\}$. We sample frames that are at least $k = 40$ frames apart.

**Implementation Details** We use ResNet-50 as the backbone following the architecture proposed in [10] for all models. We also use the same hyper-parameters as MOCOv2 [10]. In order to extract features for patches (line 10,11 of Algorithm 1 when $x_q, x_k$ are patches) in the Frame Temporal Invariance with tracks model, we use ROI-Pooling[43] at layer3 of the ResNet model. We plan to publicly release the code upon acceptance, for reproducing all the results presented in the main text.

# C   Object Part Segmentation

In Section 5 of the main text, we proposed an approach that leverages videos to learn representations using the contrastive learning framework. In this section, we perform an additional experiment to demonstrate the efficacy of our proposed model. In this experiment, we evaluate the ImageNet-based MOCO model, the ImageNet supervised model and our video-based model on the task of object-part segmentation. In order to quantify performance on this task, we use the Pascal-Parts dataset[46]. From this dataset, we crop bounding boxes for the each object and create separate images. For each object, we train a small CNN comprising of four transposed convolution layers that takes as input a single representation. In Table 5, we present the pixel-wise classification accuracies for each object and observe that our proposed model outperforms the ImageNet-MOCO model.

**Table 5: Object Part Segmentation:** We evaluate performance on the task on object-part segmentation using the Pascal Parts Dataset[46].

| Method | aeroplane | bicycle | bird | bus | car | cat | cow | dog | horse | motorbike | person | sheep | Avg. |
|---|---|---|---|---|---|---|---|---|---|---|---|---|---|
| ImageNet Supervised | 62.8 | 65.1 | 56.5 | 36.3 | 41.1 | 40.1 | 47 | 45.1 | 45.2 | 76.4 | 47.6 | 54.6 | 51.5 |
| ImageNet MOCOv2 | 57.9 | 61.2 | 49.8 | 35.5 | 41 | 39.7 | 46.9 | 44.4 | 43.7 | 75.8 | 48.8 | 51.9 | 49.7 |
| Ours Region Tracker | 62.7 | 62.2 | 51.2 | 36 | 41 | 39.9 | 47.1 | 43 | 45.2 | 75.9 | 47.2 | 53.4 | 50.4 |