[Reviews · NeurIPS 2020]

Review 1

Summary and Contributions: The paper studies the effectiveness of recently developed contrastive learning methods. It does so by building a quantitative framework for evaluating different types of invariances, suggesting the natural data augmentation of video as a method to improve performance further (with modest empirical success), and providing insight into properties of datasets that many previous methods have relied upon without mentioning explicitly.

Strengths: In my opinion, this work can be of interest to the representation learning community, since it shines light on some key aspects of the recent surge in popularity of instance-discriminative contrastive learning. Fundamentally, it makes strong points about the role of the crop in training as it relates to invariance and bias, which I found mostly convincing. The suggested approach using the natural changes in nearby video frames as a transformation is quite intuitive and makes sense, and I am pleased to see some juice squeezed out of it. To my knowledge, this is the first paper to use video data for augmentation in image representation learning.

Weaknesses: This same suggested approach seems to require using object detectors to convincingly outperform standard MOCO, and I believe that using object detectors to provide the objectness prior makes the comparison to standard MOCO somewhat unfair. Further, the continued reliance on the standard set of MOCO augmentations makes me not quite convinced of the efficacy of this method. I would have liked to see performance on just this natural frame invariance approach without the standard crop/jitter/etc augmentations which were hand designed. The argument on capacity is minimally supported by experiments and in my opinion could be toned down a bit. The now quite well known SimCLR paper (https://arxiv.org/pdf/2002.05709.pdf) shows in its Figure 1 that performance does seem to improve with # parameters, so I don't know how this squares with the authors' claims. Though the line of argument is plausible, I am skeptical of the conclusions (though they are not central to the paper anyway).

Correctness: The method is sensible and I appreciate the attempt to quantitatively evaluate networks for their learned invariances. Evaluation and training on multiple different datasets (varying the role of objects in the data) is a smart way to "demystify" these approaches.

Clarity: Overall, the paper was easy to understand and writing was fine.

Relation to Prior Work: Yes, the paper covers prior work in this fast-developing area quite well.

Reproducibility: Yes

Additional Feedback:


Review 2

Summary and Contributions: The authors propose an analysis of the invariance properties of self-supervised representation learning algorithms, and ways of inducing them. Specifically, the authors adapt the analytical framework of ref. 26, and use it to assess whether off-the-self supervised and self-supervised (MoCo and PIRL) ImageNet representations are invariant to a handful of identity preserving transformations: occlusion, viewpoint, and illumination. Although self-supervised methods are more invariant than supervised ones to occlusions, they find that supervised representations are more invariant to viewpoint and illumination. The authors then study potential biases in the evaluation of self-supervised representations. In particular, the authors ask whether the “aggressive data-augmentation” used in self-supervised training (e.g. multi-scale cropping, flipping, and color jittering) could lead to sub-optimalities that are overlooked by the current forms of evaluation. To address this question, the authors vary the “pre-training” and “testing” datasets, using either object-centric images such as ImageNet or cropped versions of PASCAL and COCO, or scene-centric datasets such as the original PASCAL and COCO datasets. They find that training and testing on the same type of data (object-centric vs scene-centric) leads to the best performance. The authors interpret these findings as an indication that “aggressive data-augmentation” leads to sub-optimal self-supervised representations. They therefore propose to additionally use the transformations occurring in natural videos as a way of generating the augmentations used in the MoCo algorithm. The authors find that the resulting representations outperform MoCo trained on static frames from the same videos, but not MoCo trained on images.

Strengths: The authors introduce many ideas and analyses in service of an ambitious goal: understanding why and under what conditions self-supervised learning leads to useful downstream representations. To that end, the authors attempt to connect several levels of description of self-supervised representations: their invariance properties, the statistics of the input data, and their downstream performance on several tasks. Connecting qualitative and quantitive analyses such as these with empirical performance is a promising and fruitful direction of research, and one which has largely been overlooked thus far. It indeed interesting that self-supervised learning is more invariant to occlusion according to the authors' analysis, and that this can be related to the augmentations used in the self-supervised training. Supervised learning often uses similar augmentations, but these need not cluster together as required by a contrastive objective. The use of cropped versions of PASCAL and COCO is also an interesting and clever way of transforming scene-centric datasets into object-centric ones. The authors present a thorough evaluation of these datasets for pre-training and downstream evaluation. Finally, the authors present an interesting way of leveraging natural videos for contrastive learning, and report results by pre-training on a dataset which had not been used for representation learning previously.

Weaknesses: The weaknesses of the paper in my mind lie in the interpretation of the results. Start with the analysis of invariances. While it is interesting that self-supervised methods are more invariant to occlusion, it is unclear why they wouldn't also be more invariant to the other augmentations used during training. For example, supervised learning appears more invariant to "Illumination Color" (Top-25 category) despite self-supervised learning methods using aggressive color augmentation techniques. This discrepancy is not discussed and we are left wondering what it means. Next, while the analysis of transfer performance as a function of cropped vs. original training and test datasets is interesting, it is unclear whether the results really support the authors' interpretation. They find that training and testing on the same type of images (i.e. object-centric, or scene-centric) results in the best performance. This is to be expected, as this minimizes the domain gap between training and testing. Instead, the authors assert that "This clearly indicates that the aggressive cropping is harmful in object discrimination and does not lead to right representation learning objective unless trained on an object-centric dataset." This is debatable: if the authors were correct, we would expect pre-training on object-centric datasets to result in the best performance across all downstream tasks, not just object-centric ones. Finally, the authors propose the use of natural videos as a way of addressing the shortcomings of the "aggressive data-augmentation" identified previously. Yet the transformations they end up using also include the MoCo v2 augmentation. Hence the total augmentation is only more aggressive, not less so. It is nevertheless interesting that these augmentations perform better than MoCo v2 augmentations in isolation, even if this deviates from their motivation and conclusions. Unfortunately, the final model still underperforms relatively to the baseline MoCo trained on ImageNet. This could be due to the downstream tasks used, but we are still left wondering whether natural videos are useful for self-supervised pre-training.

Correctness: Yes, the empirical methodology is sound.

Clarity: The paper is generally well written, if someone short on details. Some choice of notation is unfortunate: Line 158: t is used to denote an image transformation Line 170: t_i is now used to denote a threshold (in the invariance calculation) Line 179: t is again used to denote a transformation Please use separate variables for these.

Relation to Prior Work: Prior work is comprehensively discussed.

Reproducibility: Yes

Additional Feedback: POST REBUTTAL: After having read the authors' and response, I have revised my score from 4 to 7 and recommend the paper's acceptance. 1) The authors addressed the inconsistency between the additional occlusion invariance and the lack of illumination color invariance by showing how variability in illumination color is indeed not included in standard self-supervised data augmentation policies. 2) The authors also addressed the possibility that their transfer experiments (object-centric vs. scene-centric pre-training and target datasets) could simply be due to a domain-mismatch: object-centric pre-training can surpass scene-centric pre-training even on *scene-centric* target datasets given a carefully chosen subset. 3) Finally, the authors present an interesting follow-up experiment showing how video-based pre-training can surpass ImageNet pre-training for an *image-based* task. Although the task is fairly non-standard, this represents to the best of my knowledge a first, and it is important that it be documented as such.


Review 3

Summary and Contributions: This work investigates the reason behind the gains in recent self-supervised learning (SSL) methods for object detection tasks. The paper make below contributions: (1) Identifies different types of invariance that is relevant for object detection (e.g., occlusion invariance, viewpoint invariance, etc), (2) Defines an invariance metric (Top-K Representation Invariance Score) (3) Uses the above-defined metric to measure the performance of two recent SSL techniques (MOCO and PIRL) across different invariance categories, and concludes that the major gain comes from the improvement in occlusion invariance, but not the others. (4) Proposed a new SSL approach that trains a (region-tracker augmented) model on video data to improve model's ability in learning viewpoint and Illumination invariance.

Strengths: Novelty & Relevance: This paper takes an empirical step in understanding the gain in recent SSL techniques in the context of object detection. Given the recent advance in SSL, the topic is timely and relevant for the NeurIPS audience, in particular the vision community.

Weaknesses: (1) Section 4.2 could be done more rigorously. I like the two hypothesis put forward by the authors. However I do believe it is not sufficient for a NeurIPS-level publication. I strongly suggest authors to conduct additional numeric experiments to validated the two hypothesis. (2) Efficacy of using RIS score to capture invariance: The Top-K RIS seem capture invariance by measuring the firing frequency used in [26]. I struggle to understand the exact connection of this measure and the notion of invariance, i.e., the neural network's representation should admit similar distributions for the same target y. If the above definition of invariance is reasonable, then I worry that the firing rate (i.e., counting the percentage of axis that has non-zero value) only partially captures the distributional dissimilarity between different patterns. Is it possible to have two representations that are distinct from each other but their percentages of activated units are similar? Are there other alternative metrics that measure neural network invariance, and how does RIS? Some clarification on this point in Section 4.1, or some discussion on the caveat of this metric in Conclusion section should be helpful.

Correctness: I have some concern about the metric used in Equation 2 of the paper. Please see Weakness (2)

Clarity: Yes, the paper is reasonably well written. However there seems to be too much content for a conference paper. It might be helpful to integrate Section 5 as part of the empirical study, rather than a section of its own.

Relation to Prior Work: Yes. The previous work are sufficiently covered in Section 3.

Reproducibility: Yes

Additional Feedback: Line 179: "is a set a transformed versions" => "is a set of a transformed versions" === (Posted after author feedback) Thanks to authors for the feedback, I believe authors has partially addressed my concern about experiment validation. The metric, although not without its caveat, seem to be standard to the literature. Therefore I'm updating my score to 6.


Review 4

Summary and Contributions: In the context of the recent resurgence of unsupervised learning and their sometimes surprising success this paper searches for the cost that that success entails and tries a resonable first step to mitigate the shortcomings they encounter. The paper aims to answer two questions. The first is what useful invariances are built into networks trained using SOTA pretraining e.g. MOCO. They find unsuprisingly that occlusion invariance is quite strong, stronger even than using a supervised learning setup. This is not surprising given that occlusion is easily obtained in an unsupervised way using some of the oldest augmentations introduced in the literature i.e. cropping. When it comes to other useful invariances that we know humans can (mostly) rely on like viewpoint, instance, illumination dir and color this is no longer true. I think this is a very insightful and useful datapoint for the community. To achieve this they use a methodology introduced previously and the tests are performed on standard but somewhat different datasets. The quantitative evaluation is quite good I find. Secondly the paper evaluates weather the object centric nature of Imagenet (the standard pretraining dataset) played a role in invariances and the interplay with network capacity. They find that for object detection this is highly important and they seem to imply that there is some overfitting ie. higher capacity translates into better fit of the objective but because it is misaligned with some of these invariances their performance is worse too. Finally they propose to pretrain using videos instead of images to obtain the more difficult invariances for which no known random transformations can obtain data for e.g. viewpoint. They also use region proposals to obtain more instance invariant representations.

Strengths: I find the paper well written, technically sound and practically very relevant. The results should allow practitioners to reason more soundly about the practical aspects they need in their system particular as well as allow researchers to further push unsupervised learning methods and make the resulting models more broadly useful.

Weaknesses: The proposed temporal method is very promising it needs to be scaled further to obtain SOTA results on ImageNet classification. Also finding out if readily available datasets of youtube videos like kinetics are enough or something more curated is necessary still needs to be established.

Correctness: The claims are clear and quite well supported by experimental evaluations.

Clarity: yes.

Relation to Prior Work: The relation to prior work is quite well exposed.

Reproducibility: Yes

Additional Feedback: - I think it would be even more compelling to do a finetuning step to see if the gap between the invariances can be closed since keeping the representation fixed is a debatable paradigm. - longer term it would be interesting to see a large scale evaluation of the temporal model using much more pretraining data since the final performance since imagenet performance seems to still be quite a lot lower than expected.

[Author Response · NeurIPS 2020]

We thank the reviewers for their feedback. We appreciate that all the reviewers agree on the value of our analysis and they like the intuitive idea of augmentation via videos. We also acknowledge that more experiments in 4.2 (R3) would help with justification/interpretation of results (R2). We are pleased to report that we have conducted several experiments that address *all* the concerns of the reviewers. Specifically, we perform two additional experiments to highlight how aggressive augmentation hurts (and it is not just the domain-gap). We have also conducted experiments to show that on a different downstream task, *video-based augmentation outperforms even ImageNet-based MoCo*. The goal of this paper is to present a critical analysis so that the community can also introspect as we make rapid progress on the topic of representation learning. We hope the discussion here and additional experiments will convince the reviewers and AC that this message deserves a wider audience at NeurIPS.

**R2: Domain Gap or Augmentation effect? R3: Additional Experiment in 4.2** We agree: there is some ambiguity since the training data for COCO-cropped model is domain-aligned with PASCAL-cropped test data (unlike COCO-full). The drop in performance could be attributed to domain gap between COCO-full and PASCAL-cropped. So, we conducted a new experiment that cannot be explained by domain-gap.
**Setup**: Consider the subset of Pascal VOC07 images which depict either table or chair in the image, but not both (i.e scene-centric images containing only one of the frequently co-occurring pair of objects).
**Result**: On the table vs chair full image classification task, the representation trained on COCO-Boxes outperforms full COCO-image pre-training - 74.92 vs 73.64 mAP . Note that the test domain (PASCAL-full images) is aligned with training domain (COCO-full images). Yet, COCO-Boxes model outperforms COCO-full. This indicates that the problem lies with aligned representations of co-occurring objects (e.g., chairs and tables) as explained in the paper.

To support it even further, we do another experiment. We compare super-
vised learning on ImageNet with MOCO on ImageNet but for the task of
PASCAL-part classification. Note when training on ImageNet, MOCO's
cropping augmentations will learn embeddings that have similar represen-
tation for different parts. Hence, our augmentation interpretation predicts

| Method | Dog Parts | Cat Parts |
|---|---|---|
| ImageNet R50 | 69.14 | 71.31 |
| ImageNet MOCOv2 | 51.91 | 56.26 |
| (Ours) Region Tracker | 54.70 | 58.81 |

MOCO should not be as robust as supervised models on part-classifcation task. And indeed! the results indicate MOCO performance falls dramatically on part-classification.

**PIRL/MOCO should have higher Illumination Color Invariance (R2)** The natural illumination color changes for measuring invariances are restricted to illumination *temperature* changes from 2175K to 3075K - this is also naturally captured in the ImageNet dataset. The synthetically color-jitterred samples are significantly different from these images depicting arbitrary color changes (see adjoining figure). On the other hand, occlusion is very easy to accurately

Illumination Color/Temperature Changes

MOCOv2 Augmentation "ColorJitter" Samples

synthesize by simply cropping images. And therefore the cropping augmentation strategy indeed leads to high occlusion invariance even on natural images.

**Video-based Approach - Aggressive Augmentation and Performance (R1, R2)** Our method only uses the aggressive augmentation in the Frame Temporal Invariance loss. The other component of the loss involving region tracks does not employ the aggressive cropping strategy. Including the aggressive augmentations ensures that we can take advantage of the occlusion invariance that it induces, while being constrained by the region-level loss. This extra constraint ensures the best of both worlds. This method was presented as a proof-of-concept to demonstrate that the appropriate invariances can be induced by using video data. Therefore, we did not scale up the data for this approach up to be comparable to ImageNet in terms of dataset size. However, we are pleased to report (as **R2** points out) that on a downstream task which is ill-suited to ImageNet-MOCO (due to aggressive augmentation), our video approach outperforms it (even with order of magnitude less data). The results on VOC-part classification shows that trend (See table above), also demonstrating that our approach is not affected by the aggressive augmentation.

**RIS for Measuring Invariances (R3)** Goodfellow et. al. [26] explains how the firing frequency can be used to measure invariances. We directly adopt the same principle. To the best of our knowledge, this is the most relevant metric proposed in past literature. It is true that two somewhat distinct representations after thresholding can have the same invariance under this metric. However, this is an intended feature proposed in [26] since most downstream classifiers would threshold a function of the neurons (generally linear classifiers and non-linear models in some cases).

**Object Detectors as Objectness Prior (R1)**: We use a fully-unsupervised class-agnostic region proposal method - Selective Search. One should view this as a simple low-level preprocessing operation which ensures object-centric signals necessary for the augmentations. Since it uses no learning, we believe the comparisons are fair.
**Argument on Capacity (R1)**: In light of new experiments, we agree with R1 and plan to remove this as it has no connection to main story. **Finetuning representations (R4)**: For this work, we wished to analyze the representations learned in the contrastive learning framework. Recent SSL benchmarks have suggested avoiding finetuning [25] since it would cause the representations to deviate from the initially learned representations, leading to spurious inferences.

[Meta-Review · NeurIPS 2020]

The topic of the paper is very relevant to the NeurIPS community, given the increased interest in understanding self-supervised learning. Reviewers have appreciated the direction the paper takes for this, ie, to study invariances learned by self-supervised learning methods, comparing them with supervised representations. There were some concerns about the interpretations of the emprical results which have been addressed in the author response. This paper takes the first and important step towards understanding the invariances in self-supervised representations and their implications on downstream tasks, and would be of interest to the NeurIPS community. I recommend acceptance.